# A Pro-Inflammatory Stimulus versus Extensive Passaging of DITNC1 Astrocyte Cultures as Models to Study Astrogliosis

**DOI:** 10.3390/ijms25179454

**Published:** 2024-08-30

**Authors:** Leonardo A. Pérez, Esteban Palacios, María Fernanda González, Ignacio Leyton-Rivera, Samuel Martínez-Meza, Ramón Pérez-Núñez, Emanuel Jeldes, Ana María Avalos, Jorge Díaz, Lisette Leyton

**Affiliations:** 1Cellular Communication Laboratory, Center for Studies on Exercise, Metabolism and Cancer (CEMC), Instituto de Ciencias Biomédicas (ICBM), Facultad de Medicina, Universidad de Chile, Santiago 838-0453, Chilemfe.gonzalez@gmail.com (M.F.G.); ignacio.leytonrivera@gmail.com (I.L.-R.);; 2Advanced Center for Chronic Diseases (ACCDiS), Instituto de Ciencias Biomédicas, Facultad de Medicina, Universidad de Chile, Santiago 838-0453, Chile; 3Laboratorio de Microbiología Celular, Instituto de Investigación y Postgrado, Facultad de Ciencias de la Salud, Universidad Central de Chile, Santiago 833-0546, Chile; 4Andes Biotechnologies SpA, Santiago 7750000, Chile; 5Centro Científico y Tecnológico de Excelencia Ciencia y Vida, Santiago 7750000, Chile; 6Instituto de Ciencias Biomédicas, Facultad de Ciencias de la Salud, Universidad Autónoma de Chile, Santiago 7500912, Chile

**Keywords:** gliosis, reactive astrocytes, inflammation, cell passage number, α_V_β_3_ Integrin, DITNC1 cells, in vitro aging, extensive passaging, Thy-1, TNF

## Abstract

Astrogliosis is a process by which astrocytes, when exposed to inflammation, exhibit hypertrophy, motility, and elevated expression of reactivity markers such as Glial Fibrillar Acidic Protein, Vimentin, and Connexin43. Since 1999, our laboratory in Chile has been studying molecular signaling pathways associated with “gliosis” and has reported that reactive astrocytes upregulate Syndecan 4 and α_V_β_3_ Integrin, which are receptors for the neuronal glycoprotein Thy-1. Thy-1 engagement stimulates adhesion and migration of reactive astrocytes and induces neurons to retract neurites, thus hindering neuronal network repair. Reportedly, we have used DITNC1 astrocytes and neuron-like CAD cells to study signaling mechanisms activated by the Syndecan 4–α_V_β_3_ Integrin/Thy-1 interaction. Importantly, the sole overexpression of β_3_ Integrin in non-reactive astrocytes turns them into reactive cells. In vitro, extensive passaging is a simile for “aging”, and aged fibroblasts have shown β_3_ Integrin upregulation. However, it is not known if astrocytes upregulate β_3_ Integrin after successive cell passages. Here, we hypothesized that astrocytes undergoing long-term passaging increase β_3_ Integrin expression levels and behave as reactive astrocytes without needing pro-inflammatory stimuli. We used DITNC1 cells with different passage numbers to study reactivity markers using immunoblots, immunofluorescence, and astrocyte adhesion/migration assays. We also evaluated β_3_ Integrin levels by immunoblot and flow cytometry, as well as the neurotoxic effects of reactive astrocytes. Serial cell passaging mimicked the effects of inflammatory stimuli, inducing astrocyte reactivity. Indeed, in response to Thy-1, β_3_ Integrin levels, as well as cell adhesion and migration, gradually increased with multiple passages. Importantly, these long-lived astrocytes expressed and secreted factors that inhibited neurite outgrowth and caused neuronal death, just like reactive astrocytes in culture. Therefore, we describe two DITNC1 cell types: a non-reactive type that can be activated with Tumor Necrosis Factor (TNF) and another one that exhibits reactive astrocyte features even in the absence of TNF treatment. Our results emphasize the importance of passage numbers in cell behavior. Likewise, we compare the pro-inflammatory stimulus versus long-term in-plate passaging of cell cultures and introduce them as astrocyte models to study the reactivity process.

## 1. Introduction

Astrocytes, the most abundant glial cells in the central nervous system (CNS), play crucial roles in maintaining CNS homeostasis and supporting neuronal function. In healthy conditions, astrocytes display a non-reactive phenotype, but in response to brain injury or inflammation, they undergo a process known as reactive gliosis, or astrogliosis [1], triggered by pro-inflammatory cytokines [2]. Reactive astrocytes exhibit cellular hypertrophy, morphological changes, and upregulation of specific proteins, including intermediate filament proteins such as the glial fibrillary acidic protein (GFAP) and Vimentin [3]. Additionally, reactive astrocytes exhibit increased migration and proliferation rates [4,5]. Transcriptomic evidence has revealed that the transition to a reactive state entails significant changes in gene expression [6]. Our proteomic investigations have highlighted the association of astrocyte reactivity with increased levels of α_V_β_3_ Integrin, Syndecan-4, the purinergic P2X7 receptor (P2X7R), both Connexin43 (Cx43) and Pannexin1 (Px1) hemichannels (HCs) [7], as well as PI3K/AKT signaling proteins [8], all of which are crucial for different cellular responses, such as cell adhesion and migration [9,10,11].

Along with the aforementioned aspects of astrocyte reactivity, it is pertinent to acknowledge the involvement of Thy-1(CD90), a molecule abundantly expressed in neurons, which plays a significant role in modulating astrocyte adhesion and migratory responses [12,13]. Thy-1 mediates these effects through interactions with α_V_β_3_ Integrin and Syndecan-4 receptors on astrocytic surfaces [7,10,14]. However, the efficacy of Thy-1 in inducing astrocyte adhesion and migration is contingent upon the reactive state of astrocytes [7]. Reactive gliosis, which is characterized by the upregulation of various cell surface receptors and signaling molecules, enhances the responsiveness of astrocytes to Thy-1 stimulation [7,8]. Therefore, understanding the interplay between Thy-1-induced signaling and astrocyte reactivity provides valuable insights into the mechanisms underlying CNS injury and repair processes.

The immortalized rat neocortical astrocyte-derived cell line, DITNC1, has played an instrumental role in elucidating the molecular mechanisms governing astrocyte adhesion and migration triggered by Thy-1 [12,15,16]. However, research using neonatal rat primary astrocytes revealed a discrepancy: unlike DITNC1 cells, these astrocytes only responded to Thy-1 under inflammatory conditions [7,8]. This incongruity prompted us to inquire into the rationale behind the differential response patterns observed in DITNC1 cells and primary astrocytes.

Various reports have indicated that when working with cells in culture, cellular passage numbers need to be considered. For example, the increase in astrocyte passage number has been associated with the process of “aging” in vitro [17]. Using the DITNC1 cell line, these authors demonstrated that there were sudden phenotypic changes with increasing passage numbers that negatively affected astrocyte function [17]. Furthermore, an aged astrocyte model consisting of primary astrocytes has been used to compare different astrocyte phenotypes and associate them with known markers or effects of aging [18]. In vivo, the levels of GFAP and Vimentin increase with age in both rodents and humans [19]; however, the extent to which astrocytes become reactive with excessive passaging (in vitro “aging”) is unknown. Importantly, Rapisarda and coworkers found upregulated levels of β_3_ Integrin in both senescent and aging cells [20]. Of note, senescence occurs as a protective mechanism for cells facing damage and studying this process provides valuable insights into our understanding of astrogliosis. However, the relationship between in vitro passage number, β_3_ Integrin upregulation, and astrocyte reactivity has not been elucidated yet. The latter is important because of its potential to advance science in this domain while avoiding the need to use animal models.

The current investigation explored the behavioral patterns exhibited by DITNC1 astrocytes across various passage numbers and inflammatory conditions, intending to delineate their potential as a model for studying astrocyte reactivity. Simultaneously, we sought to elucidate previous disparities between DITNC1 cells and primary astrocytes. Furthermore, we explored the response of these astrocytic populations to Thy-1, serving as a functional sign of astrocyte reactivity. This study aimed to make substantive contributions to the understanding and characterization of astrocytic functionality in neuroinflammatory contexts.

## 2. Results

### 2.1. DITNC1 Astrocytes with Extensive Passaging but Not Low Passage Respond to Neuronal Thy-1 in the Absence of Inflammatory Stimuli

To test our hypothesis, we used a DITNC1 astrocyte cell line originally donated by Dr. P. Magistretti (University of Lausanne, Switzerland) in 1995. These cells, designated as DITNC1(CH), had undergone an extensive number of passages (>100). In addition, we purchased DITNC1 cells from the ATCC repository and cultured them to obtain cells with a low (LP, 3–9) and high (HP, 90–99) number of passages. The cultured LP, HP DITNC1(ATCC), and DITNC1(CH) cells differed in their monolayer disposition and cellular shape. At low passages, the cells had a moderate spindle shape and tended to form aggregates, despite having free spaces to grow attached to the plate (Figure 1A). These cells exhibited reduced spreading and prominent F-actin structures, indicative of a high level of contraction (Figure 1B). Instead, the HP DITNC1(ATCC) cells clearly aggregated and grew on top of each other, leaving free spaces between clusters (Figure 1A) with a characteristic oval shape (Figure 1B, arrow). DITNC1(CH) cells showed a spindle-like shape (Figure 1A) and exhibited a robust cytoskeleton, displaying high levels of cortical F-actin in the form of stress fibers (Figure 1B). These morphological changes suggested modified adhesion properties among these cells with a differential number of passages. Of note, for comparison, all images in Figure 1A and all of those in Figure 1B were obtained from plates seeded with similar cell numbers. To verify the astrocytic origin of all the cells used, we tested two astrocyte markers that are upregulated under pro-inflammatory conditions: C3d and S100β [19]. HP DITNC1(ATCC) and DITNC1(CH) cells showed enhanced protein levels of these markers compared to LP DITNC1(ATCC), although the values obtained were only significantly different for the DITNC1(CH) cells, while HP DITNC1(ATCC) showed intermediate values between LP DITNC1(ATCC) and DITNC1(CH) cells (Figure 1C). Given our previous reports regarding the elevated expression of β_3_ Integrin in reactive astrocytes and the fact that this is a cell adhesion molecule that could account for the variation in cell morphology observed in Figure 1A,B, we tested the total amount of this integrin in whole-cell extracts. Here, in addition to LP and HP DITNC1(ATCC) cells, we included cells with a middle number of passages [MP DITNC1(ATCC) cells, passage 36]. The levels of β_3_ Integrin gradually increased as cells went through more passages, increasing by approximately 50 and 100% with 36 and 93 passages, respectively, compared to the LP cells (Figure 1D). Importantly, the integrin levels detected in HP DITNC1(ATCC) cells were like those found in DITNC1(CH) cells (Figure 1D). Additionally, the analysis of surface β_3_ Integrin by using flow cytometry and the estimate of the median fluorescence intensity (MFI) revealed that the levels of β_3_ Integrin located at the plasma membrane increased with the passage number in DITNC1(ATCC) cells and were also elevated on the surface of DITNC1(CH) cells (Figure 1E). However, even though β_3_ Integrin levels detected in DITNC1(CH) cells were slightly higher than those found in HP DITNC1(ATCC) cells, the values obtained were not significantly different (graphs in Figure 1D,E).

Considering that both cell adhesion and β_3_ Integrin levels affect important aspects of cell behavior, such as motility [21], we tested the migration of DITNC1 cells with different passages by using the transwell assay. The neuronal glycoprotein Thy-1 was used as a migratory stimulus [14]. Of note, we have previously shown that Thy-1 induces responses in cortical primary astrocytes derived from neonatal rats only when they are reactive or have been treated with Tumor Necrosis Factor (TNF) [7]. This effect correlates with enhanced β_3_ Integrin levels, which are induced by TNF, IL-6, or IL-1β [7]. DITNC1(CH) cells migrated in response to Thy-1-Fc compared to the values obtained when treated with the negative control of the Fc protein, TRAIL-R2-Fc (Figure 1F), just as reported in most of our previous publications, and without prior TNF treatment [10,14,22]. Importantly, compared to the values obtained with TRAIL-R2-Fc, HP DITNC1(ATCC) cells migrated in response to Thy-1-Fc, whereas LP DITNC1(ATCC) cells did not (Figure 1F). Considering that our published results show that primary astrocytes only migrate after Thy-1 stimulation when previously exposed to an inflammatory stimulus and that they upregulate surface and total levels of β_3_ Integrin [7,23], we suggest that Thy-1-induced migration of HP DITNC1(ATCC) cells not treated with TNF (Figure 1F) might be due to increased expression of β_3_ Integrin (Figure 1D,E).

Altogether, these results indicate that long-term cultures of DITNC1 astrocytes undergo changes in cell morphology and β_3_ Integrin expression levels. In addition, these astrocytes with extensive passaging show elevated β_3_ Integrin at the cell surface and migrate when exposed to Thy-1 without requiring a pro-inflammatory cytokine treatment.

### 2.2. Cells with a Low Number of Passages Behave as Non-Reactive Astrocytes and Switch to Reactive Astrocytes When Treated with TNF

Building upon the above findings, which correlated an increased number of passages with the effects of an inflammatory stimulus, our prediction was that the LP DITNC1(ATCC) cells behaved like non-reactive astrocytes. If this were the case, then under TNF treatment, LP DITNC1(ATCC) cells would become reactive and respond to Thy-1, just as reported for TNF-treated rat primary astrocytes and the DITNC1(CH) cells [7,10,14,22,23].

To test this idea, we incubated LP DITNC1(ATCC) cells with TNF and checked for reactivity markers by both immunoblot and immunofluorescence as well as functions changed in reactive astrocytes such as adhesion, migration, and proliferation (Figure 2A). The classical reactivity marker is GFAP. When incubated for 24, 48, and 72 h, these cells showed elevated GFAP expression levels by immunoblotting, with a peak at 48 h (Figure 2B), as previously described in rat primary astrocytes [7]. Then, using LP DITNC1(ATCC) cells incubated with TNF for 48 h, we tested for other reactivity markers, such as Cx43 and β_3_ Integrin. As shown in Figure 2C, both markers were upregulated with TNF treatment when assessed by immunoblots. In addition, another recognized reactivity marker, Vimentin [6], as well as Cx43, were enhanced by TNF treatment, as revealed by immunofluorescence staining (Figure 2D). Moreover, TNF induced changes in the subcellular localization of Cx43 to the perinuclear compartment and the cell border (Figure 2D).

Functional features of reactive astrocytes were also tested. We first assessed FA formation by immunofluorescence. We examined LP DITNC1(ATCC) cells under two conditions: untreated (Control) and treated with TNF (dashed gray rectangle in Figure 2E). Subsequently, these cells were stimulated with Thy-1 or treated with the negative control, TRAIL-R2. Our analysis revealed that only those LP DITNC1(ATCC) cells treated with TNF exhibited a robust response to Thy-1 (Figure 2E), leading to a significant increase in the number of FAs per cell comparable to that observed in DITNC1(CH) cells treated with Thy-1 (Figure 2F). Interestingly, the levels of FA/cell were elevated in TNF-treated LP DITNC1(ATCC) cells, with TRAIL-R2 treatment surpassing basal control levels and resembling those observed in DITNC1(CH) cells under similar conditions (Figure 2F). We then tested Thy-1-induced astrocyte migration. Here, results indicated that LP DITNC1(ATCC) cells migrated in response to Thy-1 only when treated with TNF but not when exposed to the control TRAIL-R2 (Figure 2G). On the other hand, DITNC1(CH) cells treated with Thy-1 migrated regardless of prior TNF treatment (Figure 2H). To test proliferation, we used the MTS assay. Here, TNF-treated LP DITNC1(ATCC) and DITNC1(CH) cells significantly increased their proliferation at 72 h, compared to LP DITNC1(ATCC) without TNF treatment at the same time point (Figure 2I). However, the values were not significantly different when comparing the absorbance (490 nm) of different cell types ± TNF at either 24 or 48 h (Figure 2I). DITNC1(CH) cells proliferated at a similar rate, independent of the presence of TNF (Appendix A). In addition, both cell types, treated or not with TNF, exhibited increased proliferation at 72 h compared to 24 h and 48 h (Appendix A). Therefore, LP DITNC1(ATCC) cells treated with TNF adopt reactive astrocyte characteristics, including increased adhesion, migration, and proliferation. These observations shed light on the differential responses of astrocytes to Thy-1 stimulation, dependent on their reactive status and pretreatment with TNF. Furthermore, the results confirm that DITNC1(CH) cells behave like reactive astrocytes without needing any inflammatory stimuli.

### 2.3. Reactive Phenotype Traits of DITNC1 Cells with Extensive Passaging or Treated with Pro-Inflammatory Stimuli Are Not Due to Senescence

Upon brain injury, astrocytes take two alternative pathways in an inflammatory environment: senescence or astrogliosis [24]. Cellular senescence shares many features with astrogliosis, such as cellular hypertrophy and the secretion of pro-inflammatory molecules [24]. Our previous assays with DITNC1(ATCC) cells revealed that multiple passages [i.e., HP DITNC1(ATCC) cells] led to increased β_3_ Integrin expression and made these cells Thy-1 responsive, even in the absence of pro-inflammatory stimuli (Figure 1). These findings align with established indicators of astrogliosis [7], although they may also suggest potential senescent attributes [20]. To test a possible role for senescence, we assessed senescence-associated β-galactosidase (SA-β-galactosidase) activity (Figure 3A) and investigated cell cycle arrest (Figure 3B), both recognized hallmarks of senescent cell populations [25]. No differences were found in these two parameters in astrocytes with different passages. Neither hydrogen peroxide (H_2_O_2_) nor TNF treatments elicited alterations in SA-β-galactosidase activity in astrocytes. On the other hand, cervical squamous carcinoma cells (SiHa), serving as a positive control, exhibited increased β-galactosidase activity when exposed to H_2_O_2_ (Figure 3A). SiHa cells have a notorious sensitivity to H_2_O_2_-induced oxidative stress, resulting in escalated reactive oxygen species levels and suppressed cell proliferation rates [26]. Additionally, no significant changes were observed in the proportion of cells in the G0/G1 phase, and all cell populations showed a similar pattern of cell cycle phases (Figure 3B). These results allowed us to discard the potential involvement of senescence pathways in the observed alterations, underscoring the complex interplay between cell passages and astrogliosis within astrocyte populations.

These results highlight the importance of the passage number in cell behavior and foresee the LP DITNC1(ATCC) cells as a model for non-reactive astrocytes. Specifically, we demonstrate that LP DITNC1(ATCC) cells serve as a valuable model for non-reactive astrocytes under basal conditions. However, these cells exhibited the capacity to transition to a reactive phenotype in response to inflammatory stimuli, accompanied by increases in the expression of reactivity markers, cell adhesion, migration, and proliferation. Additionally, the DITNC1(CH) cells, used in most of our previous studies [14,22], behaved as reactive astrocytes and did not require TNF to respond to Thy-1. Furthermore, DITNC1(ATCC) cells with multiple passages (HP) did not show a senescent phenotype but presented traits of cells with a reactive phenotype.

### 2.4. DITNC1 Astrocytes with Multiple Passages Inhibit Neurite Outgrowth and Promote Neuronal Death

One of the most important aspects of reactive astrocytes is their effect on neurons. Our previous reports have shown that DITNC1(CH) cells inhibit neurite outgrowth due to the interaction of α_V_β_3_ Integrin in astrocytes with Thy-1 in neurons [27,28,29]. Here, we separated the effect of membrane proteins from that of secreted factors by fixing the astrocytes in the plate (Figure 4A). Using this approach, we show that LP DITNC1(ATCC) cells only affected neurite length when treated with TNF, just like neurons differentiated on DITNC1(CH) in the absence of TNF (Figure 4B,C). Neurons differentiated over both LP DITNC1(ATCC) astrocytes with TNF and DITNC1(CH) astrocytes without TNF exhibited shorter processes than control LP DITNC1(ATCC) without TNF (Figure 4B,C). Additionally, the lengths of neurites grown over LP DITNC1(ATCC) cells were not significantly different from those of neurons differentiated over the plastic surface of a plate (Figure 4B,C).

The principal role of astrocytes is to maintain homeostasis in the CNS: to protect synapses, regulate neuronal signaling, and protect neurons from damage [30]. Conversely, after brain injury or under inflammatory conditions, astrocytes undergo astrogliosis, switch to a reactive phenotype, and secrete factors that exert toxic effects on neurons [1]. Hence, we evaluated if the astrocyte-conditioned media (ACM) from the LP DITNC1(ATCC) cells treated with TNF were able to generate a toxic effect on neurons (Figure 4A), as we reported for the DITNC1(CH) cells [27,28,29]. Our results show that the ACM from LP DITNC1(ATCC) cells was detrimental for a small number of cells (20%), just as reported for ACM from primary astrocytes [31]; however, when the LP DITNC1(ATCC) cells were treated with TNF, a significant number of neurons died (40%), compared to the control without TNF (Figure 4D). Likewise, the ACM obtained from DITNC1(CH) cells significantly increased the number of dead neurons (Figure 4D). Altogether, these results suggest that LP DITNC1(ATCC) cells treated with TNF and DITNC1(CH) cells (without prior TNF treatment) secrete neurotoxic factors, just like primary astrocytes with a reactive phenotype [32,33,34].

## 3. Discussion

Due to their important role in maintaining brain homeostasis, astrocytes are considered key players in neurological disorders. Astrocytes undergo physiological state changes in response to injury, neurodegeneration, and aging as they respond to inflammation by turning into a reactive neuroprotective (A2-type) or neurotoxic (A1-type) stage [30,35]. Astrocytes from aged mice are reactive and neurotoxic, making them unable to comply with normal astrocyte function [19]. Here, we show that long-term cultured DITNC1 astrocyte cell lines undergo changes in cell morphology, show more β_3_ Integrin at the cell surface, and migrate when exposed to Thy-1. In addition, LP DITNC1(ATCC) cells, with only a few passages after ATCC reception, behave as non-reactive astrocytes. However, they show signs of reactivity, increased adhesion, migration, and proliferation when treated with a pro-inflammatory stimulus such as TNF. Therefore, we describe two models to study astrocyte reactivity: (1) non-reactive astrocytes [DITNC1(ATCC), with low passage number] that can become reactive under inflammatory stimuli, and (2) reactive cells [DITNC1(CH)] that behave as reactive astrocytes in the absence of TNF but exhibit reactive astrocyte traits and respond to Thy-1. The first model is valuable for understanding the transition from a normal to a reactive astrocyte state, particularly in response to specific inflammatory triggers. The second model, on the other hand, provides insights into the inherent properties and responses of reactive astrocytes without external inflammatory stimuli. The significance of each model lies in their complementary roles. LP DITNC1(ATCC) serves as a controlled system to study the induction of reactivity, while DITNC1(CH) provides a model for studying inherent reactive traits. By comparing these models, we gain a comprehensive understanding of both the mechanisms underlying astrocyte reactivity and the impact of cell culture conditions on astrocyte behavior. These models are crucial for exploring how astrocyte reactivity evolves and contributes to neuroinflammatory conditions.

Noteworthy, DITNC1(ATCC) cells that undergo many passages (>90) display various traits of reactive astrocytes. This indicates that extensive passaging of the cells in culture has a profound effect on astrocyte functions, resembling cells with a reactive phenotype, like that of DITNC1(CH) cells. This evidence demonstrating that multiple passages modify astrocyte reactivity is vital, as it highlights the potential challenges and limitations associated with the use of long-term cultures. Beyond proving the different models, our findings open several future research avenues. The need to develop protocols or standards that closely control the effects of passaging is a significant takeaway, thereby ensuring the physiological relevance of in vitro models. Furthermore, we could apply the methodology of the present study to systematically investigate passaging-induced changes in other cell lines. For instance, endothelial cells, known to be activated by inflammatory stimuli [11], may also undergo activation solely due to prolonged passaging. In addition, these insights can contribute to the refinement of in vitro models, making them more accurate predictors of in vivo responses and enhancing translational research in the field of inflammation.

Astrocytes are classified as naïve (non-reactive), reactive, or scar-forming astrocytes based on their markers and location. Reactive astrocytes express β-catenin and are motile, whereas naïve and scar-forming astrocytes express N-Cadherin and are not motile [36]. These reported findings suggest that there is a temporal sequence in the progression from naïve to reactive and then to scar-forming astrocytes, with only reactive astrocytes being motile. Acute injury, inflammation, and neurodegenerative diseases (such as Alzheimer’s, Amyotrophic Lateral Sclerosis, and ALS) can all cause pro-inflammatory conditions that can convert naïve astrocytes into motile and reactive cells [19]. We have reported in vitro studies that have also shown that primary astrocytes from neonatal rats migrate in response to neuronal Thy-1 only after being treated with TNF, which induces their reactivity [7]. Moreover, reactive astrocytes corresponding to TNF-treated cells or derived from a transgenic mouse model of ALS (hSOD^G93A^) exhibit increased levels of reactivity markers (GFAP, Cx43, and β_3_ Integrin), and proteins involved in astrocyte adhesion and migration [7]. Thus far, Thy-1 is the only protein in neurons reported to bind to α_V_β_3_ Integrin and promote astrocyte migration [16,37]. Therefore, considering the results obtained with the different astrocyte cell lines used in this study, LP DITNC1(ATCC) treated with TNF, HP DITNC1(ATCC), and DITNC1(CH) show reactive astrocyte traits given the upregulation of α_V_β_3_ Integrin and their migratory properties.

NF-κB-dependent pathways, linked to several neurodegenerative diseases, turn astrocytes into the reactive and neurotoxic A1-type [38]. Using primary astrocytes, we recently reported that TNF activates NF-κB, increasing the expression of GFAP, Vimentin, Cx43, and α_V_β_3_ Integrin [31]. Furthermore, the conditioned medium from these reactive astrocytes harms neurons, suggesting that TNF induces a neurotoxic A1 phenotype [31]. In the present study, we show that LP DITNC1(ATCC) treated with TNF and DITNC1(CH) correspond to reactive astrocytes likely of the A1-type, capable of damaging neurons through membrane-expressed proteins or factors released to the medium (ACM). Therefore, TNF-induced astrocyte reactivity, along with factors secreted by these reactive DITNC1 astrocytes, contribute to neuronal damage.

Reactive astrocytes reportedly secrete neurotoxic factors, including ATP and polyphosphates, which may be key cytotoxic components in the conditioned medium. ATP, released from damaged cells, binds to purinergic receptors on neurons, triggering calcium influx and detrimental signaling pathways [39]. Similarly, polyphosphates interact with neuronal membranes and receptors, exacerbating calcium dysregulation and inflammation [34]. These molecules collectively inhibit neurite outgrowth and induce neuronal death by activating inflammatory cascades and producing reactive oxygen species [34]. Thus, ATP and polyphosphates are likely critical factors contributing to the observed neurotoxicity in the conditioned medium from DITNC1 reactive astrocytes; however, further investigations are required to identify the toxic factors present in the ACM of LP DITNC1(ATCC) treated with TNF and DITNC1(CH) cells.

Aging is characterized by chronic but low-intensity systemic inflammation, and a comparative transcriptomic analysis of young versus aged mouse astrocytes showed that the latter exhibited upregulated gene expression, affecting astrocyte reactivity, synapse elimination, and immune response pathways [19]. Remarkably, our present study shows that astrocyte reactivity is reproduced by extensive passaging in culture, a process that could resemble that of chronological aging in vivo; however, in vitro studies could only suggest similarities rather than causative evidence of the process. Aged astrocytes are neurotoxic and express β_3_ Integrin [24], just as our reactive astrocyte cell models. We have previously shown that by silencing β_3_ Integrin, TNF does not influence astrocyte phenotype; however, after overexpressing this integrin, astrocytes display many features of reactive astrocytes in the absence of any pro-inflammatory treatment [7]. In a Appendix A of this same study by Lagos-Cabré, we showed that other cytokine treatments, like IL-6, IL-1β, and IFNγ, also enhance the expression of α_V_β_3_ Integrin in primary astrocytes [7]. Therefore, we propose identifying β_3_ Integrin in astrocytes as a relevant reactivity marker for the study of neurodegenerative processes. In this regard, it would also be interesting to study whether the aged astrocyte phenotype can be reverted by silencing β_3_ Integrin.

Our study agrees with others, stating that cell passages in culture can mimic the aging process. However, it is important to exercise caution since chronological aging involves various biological, molecular, and biochemical changes that do not necessarily correspond to those induced by extensive in vitro passaging. The referred studies revise diverse astrocyte functions, which are different from those shown in the present study. For example, astrocytes with high passage numbers show compromised mitochondrial membrane potential, decreased neuroprotective functions, more pronounced ATP-stimulated Ca^2+^ signaling [40,41,42,43], and nuclear enlargement [44]. Of note, the DITNC1 model of reactive astrocytes proliferates and does not exhibit signs of senescence, features that primary astrocytes with seven passages are shown to display [18]. Therefore, our report confirms and expands the fact that astrocytes in cultures with extensive passaging can be cautiously used as a model for in vitro-aged and reactive astrocytes.

Astrocytes with a reactive phenotype are crucial in traumatic events, inflammatory processes, and most neurodegenerative diseases. Therefore, in vitro models are needed to study the features of these astrocytes. The LP DITNC1(ATCC) model provides a controlled environment to study astrocyte-induced reactivity, allowing for systematic exploration of specific mechanisms involved in neuroinflammation. It enables precise analysis of how stimuli or genetic alterations might influence reactive astrocytes. It might also help elucidate how reactive astrocytes contribute to disease progression in conditions such as Alzheimer’s and multiple sclerosis by modeling how reactivity impacts neural networks and inflammatory pathways. This controlled environment is ideal for detailed mechanistic studies. In contrast, the DITNC1(CH) model reflects the inherent reactive traits of astrocytes. It captures baseline reactivity and responses in astrocytes without additional external inducements, which can reveal insights into the fundamental traits and variability in astrocyte behavior in various inflammatory conditions. It can also be used to investigate the potential impact of astrocyte reactivity variations on disease susceptibility and the progression of conditions such as chronic neurodegeneration. Therefore, despite acknowledging the inherent limitations of in vitro models, these systems represent significant advancements in studying astrocyte responses under inflammatory conditions.

## 4. Materials and Methods

### 4.1. Cell Culture

DITNC1 cells [15] were obtained from the American Type Culture Collection (ATCC), and DITNC1(CH) cells were kindly donated by Dr. P. Magistretti (University of Lausanne, Lausanne, Switzerland). These cells were cultured in RPMI (Gibco Invitrogen, Life Technologies, Carlsbad, CA, USA) with the antibiotics 100 UI/mL ampicillin and 100 µg/mL streptomycin (Sigma-Aldrich, St. Louis, MO, USA) and 5% Fetal Bovine Serum (FBS, Hyclone, Pittsburgh, PA, USA) (complete medium, CM). SiHa cells from ATCC (CCL-2) were cultured in Dulbecco’s modified Eagle’s medium (DMEM High GlutaMax; Gibco Invitrogen, Life Technologies) containing 10% FBS and antibiotic–antimycotic (Gibco Invitrogen, Life Technologies). The catecholaminergic neuronal-like cell line (CAD cells) [45] was cultured in DMEM-F12 (Gibco Invitrogen, Life Technologies) media, supplemented with 8% FBS and antibiotics. All these cells were cultured in a humidified atmosphere with 5% CO_2_ at 37 °C.

### 4.2. Reagents and Antibodies

The Thy-1-Fc and TRAIL-R2-Fc proteins were made in Switzerland and kindly provided by Dr. Pascal Schneider (University of Lausanne, Epalinges, Switzerland) [46]. For cell stimulation, the fusion proteins were coupled to protein A, or protein A/sepharose beads (Sigma-Aldrich) in a 10:1 ratio (Fc fusion protein-protein A) and incubated at 4 °C for 1 h. Bovine serum albumin (BSA) and sodium selenite were from Sigma-Aldrich. Immunofluorescence reagents from Sigma-Aldrich were rhodamine-conjugated phalloidin, DAPI (diamidino-2-phenylindole), and mouse anti-vinculin mAb. Other antibodies were goat anti-mouse IgG conjugated to Alexa fluor 488 (Molecular Probes, Eugene, OR, USA), rabbit anti-β_3_ Integrin pAb (Millipore, Billerica, MA, USA), mouse anti-α_V_β_3_ Integrin/Phycoerythrin (PE-A, Santa Cruz Biotechnology, Dallas, TX, USA), rabbit anti-Syndecan-4 pAb (Abbexa, Cambridge, UK), mouse anti-Cx43 mAb (Thermo Fisher Scientific, Waltham, MA, USA), mouse anti-HSP90 α/β mAb, mouse anti-Vimentin mAb (Santa Cruz Biotechnology), rabbit anti-C3d mAb (Abcam, Cambridge, MA, USA), rabbit recombinant anti-S100β mAb (Abcam), and mouse anti-GFAP mAb (Sigma-Aldrich). Horseradish peroxidase (HRP)-conjugated goat anti-mouse IgG pAb (Bio-Rad Laboratories, Inc., Hercules, CA, USA) or goat anti-rabbit IgG-HRP pAb (Jackson ImmunoResearch Labs, Inc., West Grove, PA, USA) were used for immunoblotting.

### 4.3. Immunoblot Analysis

Whole-cell lysates were obtained from rat DITNC1 astrocytes in an ice-cold lysis buffer, as previously described [10]. Protein extracts (50 μg/lane) were run on SDS-PAGE (10% gels) and transferred to nitrocellulose membranes. Membranes were blocked with 5% BSA or 5% low-fat milk in PBS and then incubated with anti-β_3_ Integrin, anti-Cx43, anti-C3d, anti-S100β, anti-GFAP, anti-HSP90, or anti-β actin, followed by a second antibody coupled to HRP. Peroxidase activity was detected as enhanced chemiluminescence using a digital Hamamatsu camera (Hamamatsu Photonics K.K., System Division, Hamamatsu City, Japan). Densitometric analysis of the bands obtained was quantified using the ImageJ program.

### 4.4. Cell Cytometry

Trypsin-detached astrocytes were incubated and processed at 4 °C to avoid the internalization of surface proteins. After blocking with 5% BSA, the cells were incubated with anti-α_V_β_3_ Integrin/Phycoerythrin (PE-A) for 2 h. Cells were washed with PBS and analyzed using a FACS Canto flow cytometer (BD Bioscience, Franklin Lakes, NJ, USA) to measure integrin surface levels in astrocytes. Data were analyzed and plotted using the FlowJo software (version Vx).

### 4.5. Migration Assay in a Boyden Chamber

Migration assays were performed in Boyden chambers (Transwell Corning, 6.5 mm diameter, 8 µm pore size, Kennebunk, ME, USA) as reported [47]. The bottom side of the inserts was coated with 2 mg/mL of fibronectin. Astrocytes were treated with 10 ng/mL TNF for 48 h, where indicated, and then 100,000 cells per condition were plated on the top insert in serum-free medium (SFM). The complete medium, which acts as a chemoattractant, was added to the bottom chamber. After 2 h of stimulation with Thy-1-Fc or with TRAIL-R2-Fc added to the top chambers in SFM, inserts were washed, the cells on the top of the inserts were removed with cotton swabs, and those at the bottom side were stained with 0.1% Crystal Violet in 2% ethanol. Migrating astrocytes were then counted under an inverted microscope. At least six fields per condition were evaluated in each independent experiment.

### 4.6. Indirect Immunofluorescence Assay for Visualization of Focal Adhesions

DITNC1 astrocytes were grown on sterile coverslips in 24-well plates and, where indicated, treated with 10 ng/mL of TNF for 48 h in complete medium. The cells were stimulated with Thy-1-Fc:Protein A (4 μg:0.4 μg)/sepharose beads in SFM for 15 min. SFM and TRAIL-R2-Fc-Protein A were used as negative controls. Astrocytes were fixed with 4% p-formaldehyde in fixing buffer (100 mM PIPES, pH 6.8, containing 2 mM EGTA, 2 mM MgCl_2_, and 0.04 M KOH). Cells were then washed three times with universal buffer (50 mM Tris-HCl, pH 7.6, 0.15 N NaCl, and 0.1% sodium azide) at room temperature for 15 min. The cells were permeabilized with 0.1% Triton X-100 in the universal buffer for 10 min, washed twice with the same buffer without detergent, and then blocked with 2% BSA. FAs were identified with the mouse anti-vinculin mAb, actin stress fibers with rhodamine-conjugated phalloidin, and DAPI was used to stain the nuclei. Fluorophores were visualized in an Olympus IX81 DSU microscope (Olympus, Center Valley, PA, USA) using a PLAPON 40× objective with the XM10 camera. The quantification of FA number per cell was performed by selecting structures in a determined pixel range above a threshold value. We used the ImageJ “Analyze particle” plugin, as previously described by our lab [14].

### 4.7. Proliferation Assays

DITNC1 astrocytes (2 × 10^3^/well) were seeded in 96-well plates. Cells were treated for 24, 48 h, and 72 h with TNF (10 ng/mL). The cell number was evaluated by adding fresh culture medium containing 10% of the compound 3-(4,5-dimethylthiazol-2-yl)-5-(3-carboxymethoxyphenyl)-2-(4-sulfophenyl)-2H-tetrazolium (MTS) of the CellTiter 96^®^ aqueous non-radioactive cell proliferation assay, according to the instructions specified by the manufacturer (Promega, Madison, WI, USA) [48]. Soluble formazan produced by viable cells was detected by measuring the absorbance at 490 nm on a SPECTROstar Nano microplate reader. Background values corresponding to excess cell debris and bubbles were subtracted by measuring the absorbance at 650 nm. PBS was used as the vehicle.

### 4.8. Beta-Galactosidase Senescence Assay

Senescence was determined by staining senescence-associated-β-galactosidase (SA-β-galactosidase) activity with the Senescence Detection Kit (Abcam), according to the manufacturer’s specifications. DITNC1 or SiHa cells (3 × 10^3^/well) were seeded in 4-well plates (Nunc, Invitrogen, Waltham, MA, USA) and cultured for 24 h. Cells were then left untreated or treated with 10 ng/mL of TNF or 10 µM H_2_O_2_ for 48 h. SiHa cells treated with H_2_O_2_ were used as SA-β-galactosidase-positive control cells [49]. After treatment with TNF or H_2_O_2_, the cells were washed with PBS and incubated in the kit’s fixative solution for 15 min. at room temperature. Each well was then gently washed twice with PBS and incubated with the kit’s staining solution for 18 h at 37 °C in a humidified chamber. The cells were then washed with PBS and analyzed under a fluorescent and phase-contrast Olympus YX71 inverted microscope.

### 4.9. Cell Cycle Assay

DITNC1 astrocytes (5 × 10^4^/well) were seeded in the complete medium and left to adhere for 24 h in 24-well plates. Then, DITNC1 astrocytes were treated with TNF (10 ng/mL) for 48 h. Cells were fixed and permeabilized with cold methanol (−20 °C) for 10 min. Subsequently, astrocytes were resuspended with RNAse (100 μg/mL) in PBS and stained with propidium iodide (10 μg/mL), which binds to DNA, to determine the cell cycle stages (G0/G1, S, and G2/M). Propidium iodide has an excitation peak of 550 nm and an emission peak of 620 nm, which was detected by a FACSCanto BD flow cytometer. The fluorescence detected correlates with the amount of DNA in the cells. The WinMDI computer program was used to perform the analysis.

### 4.10. Outgrowth of Neuronal Processes on Fixed Astrocyte Monolayers

Astrocytes were seeded on glass-bottom 6-well plates at a density of 5 × 10^5^ cells/cm^2^ and incubated overnight. Then, astrocytes were treated with or without TNF (10 ng/mL) for 48 h. Next, the medium was eliminated, and the cells were washed twice with PBS, fixed with 4% p-formaldehyde in fixing buffer for 15 min at room temperature, and washed with 0.1 mM glycine and abundant PBS. Then, CAD cells labeled with Cell Tracker Green CMFDA (10 µM) were seeded on a plate or on fixed astrocytes at a density of 1 × 10^4^ cell/cm^2^ in DMEM-F12 media supplemented with 8% FBS and antibiotics [27]. After culturing overnight, the media was replaced with fresh SFM containing 50 ng/mL sodium selenite to induce neuronal morphological differentiation for 24 h. After this time, neuronal processes were captured with a Cytation 3 instrument (BioTek, Santa Clara, CA, USA) or an epifluorescence spinning disk microscope (Olympus). Quantification of the length of neurites was determined using the NeuronJ plugin of the NIH ImageJ software v1.8, as previously described [28,29].

### 4.11. Neuronal Cell Death Assay

DITNC1 astrocytes (5 × 10^4^/well) were seeded in the complete medium and left to adhere for 24 h in 24-well plates. Then, DITNC1(ATCC) astrocytes were treated with TNF (10 ng/mL) for 48 h or left untreated. DITNC1(CH) astrocytes were incubated for the same time without treatment. Then, the culture medium was removed and changed to serum-free DMEM-F12 medium to rule out the participation of TNF used for the treatment of LP DITNC1(ATCC) astrocytes. The astrocyte-conditioned medium (ACM) was obtained by incubating these astrocytes for 4–5 days in the freshly added medium. Subsequently, the ACM was collected, filtered, and used to test for possible toxic effects on CAD cells. To this end, CAD cells were differentiated for 24 h in SFM containing 50 ng/mL of sodium selenite. Then, differentiated CAD cells were treated with ACM for 24 h. Finally, the cells were washed with PBS, detached, transferred to cytometer tubes, and incubated with propidium iodide (10 µg/mL). Cells with propidium iodide indicated the cell death/permeability of the cells.

### 4.12. Statistical Analysis

The data are expressed as the mean ± standard error of the mean (s.e.m.) of results from three independent experiments (n = 3). Results were analyzed using the non-parametric Mann–Whitney test to compare two groups and the Kruskal–Wallis post-test to compare multiple groups. A probabilistic value of *p* < 0.05 was considered to be significantly different. The statistical analysis was performed using GraphPad Prism software (version 10, Ashland, OR, USA).

## 5. Conclusions

The long-lived DITNC1 astrocyte cell line represents a model of reactive astrocytes. DITNC1 cells with only a few passages after ATCC reception behave as non-reactive astrocytes, and when treated with a pro-inflammatory stimulus (such as TNF), show reactivity traits, undergo cell morphology changes, show more β_3_ Integrin at the cell surface, and increase adhesion and migration properties when exposed to the neuronal Thy-1 protein. Therefore, DITNC1(ATCC) cells with low passage numbers represent a model of non-reactive astrocytes that can be used to study astrocyte reactivity when exposed to inflammatory stimuli. Noteworthy, DITNC1 cells undergoing multiple passages (>90) present various traits of reactive astrocytes, indicating that extensive passaging of the cells in culture has a profound effect on astrocyte functions. In addition, these cells could serve as a model to study astrocyte reactivity. While in vitro models lack the complexity of in vivo systems, their combined use advances our understanding of astrocyte reactivity both mechanistically and physiologically. Integrating advanced techniques like high-throughput screening and multi-omics could further enhance the value of these models, allowing for deeper insights into neuroinflammatory diseases and more accurate predictions of disease progression. These approaches could speed up the discovery of new treatments targeting reactive astrocytes while also expanding our knowledge of the biology of these conditions and diseases, ultimately helping to improve treatments for neuroinflammatory and neurodegenerative diseases.

## Figures and Tables

**Figure 1 ijms-25-09454-f001:**
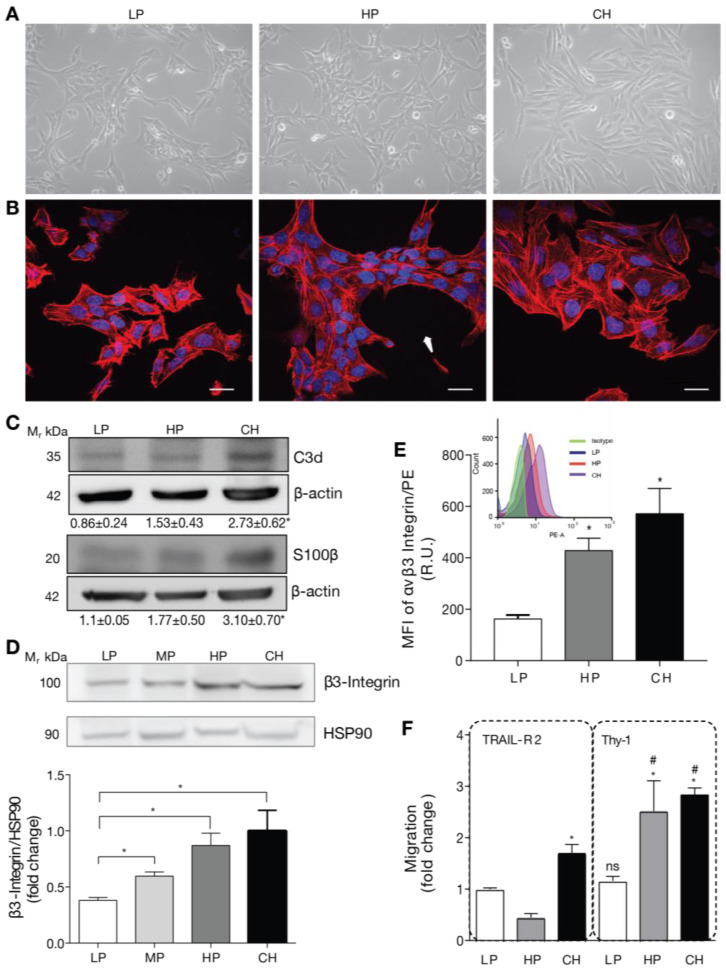
DITNC1 astrocytes with extensive passaging but not low passages respond to neuronal Thy-1 without inflammatory stimuli. (**A**) Phase contrast microphotographs of low passage (LP), high passage (HP), DITNC1(ATCC), and DITNC1(CH) astrocytes grown for 48 h. Magnification = 200×. (**B**) Immunofluorescence microphotographs obtained with the confocal microscope of LP, HP DITNC1(ATCC), and DITNC1(CH) astrocytes grown for 48 h and stained with DAPI (nuclei, blue) and rhodamine-labelled phalloidin (F-actin, red). Cells clustered leaving free spaces in the plate with a characteristic oval shape (white arrow). Scale bar = 20 µm. (**C**) LP, HP DITNC1(ATCC), and DITNC1(CH) astrocyte lysates were immunoblotted with antibodies against C3d or S100β and β-actin as a loading control. Densitometric and statistical analyses are shown below each band as the mean ± s.e.m. (**D**) LP, middle passage (MP), HP DITNC1(ATCC), and DITNC1(CH) astrocyte lysates were immunoblotted with antibodies against β_3_ Integrin and HSP90 as a loading control. This graph shows the averaged β_3_ Integrin band intensity normalized to HSP90, the values expressed as ratios, and the fold-change of the mean LP value. (**E**) Insert: A histogram obtained by flow cytometry to evaluate β_3_ Integrin levels on the surface of astrocytes. The three cell populations correspond to the LP (blue), HP (orange), and CH (purple) cells, and the isotype control is shown (green). Graph: median fluorescence intensity (MFI) determined by flow cytometry, shown in the histogram (insert). Bars in the graph indicate the values obtained from LP (white), HP (dark gray), and CH (black) cells. (**F**) The migration assay of LP, HP DITNC1(ATCC), and DITNC1(CH) astrocytes was performed as described in Materials and methods. All the cell lines were stimulated with Thy-1-Fc conjugated to Protein A [Thy-1-Fc/Protein-A (4 µg/0.4 µg per well)] for 2 h or with TRAIL-R2-Fc as a negative control. The graph shows values normalized to the average of LP DITNC1(ATCC) astrocytes under the control condition. Values in (**D**–**F**) correspond to the mean ± s.e.m (n = 3). Significant differences are indicated. * *p* < 0.05 compared to LP values [or LP/TRAIL-R2 in (F)]. # *p* < 0.05 compared to each respective control treated with TRAIL-R2. R.U = Relative Units.

**Figure 2 ijms-25-09454-f002:**
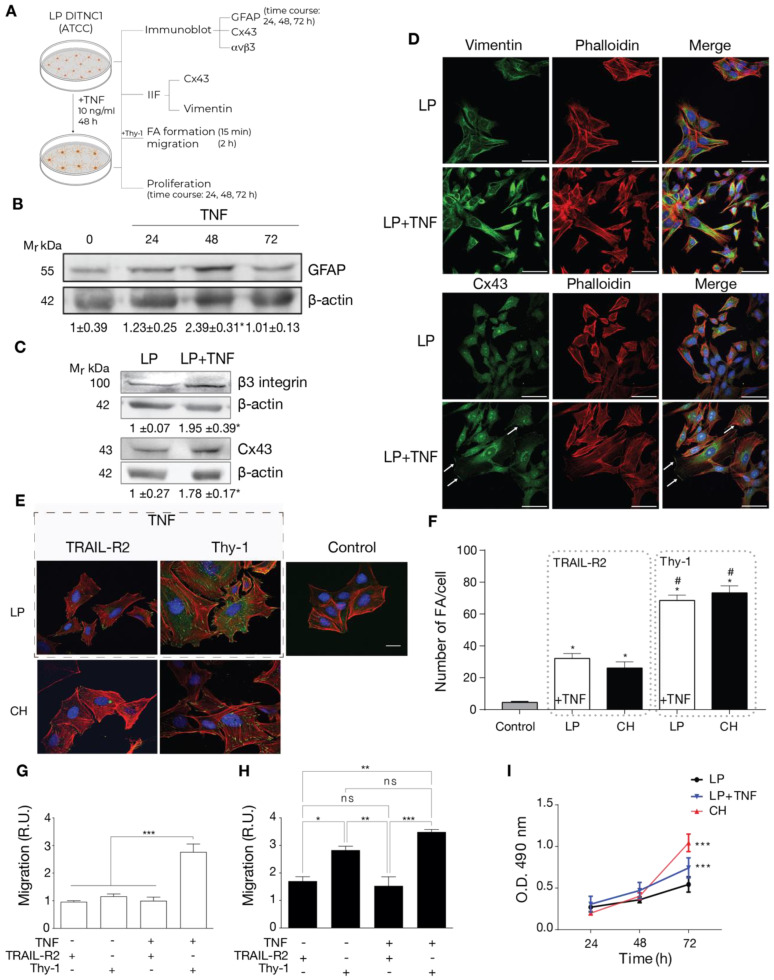
Cells with a low number of passages behave as non-reactive astrocytes and switch to reactive astrocytes when treated with TNF. (**A**) LP DITNC1(ATCC) astrocytes treated with TNF (10 ng/mL) for 48 h (or the indicated periods) or left untreated were subjected to immunoblots, immunofluorescence, cell adhesion, migration, and proliferation assays. (**B**) Immunoblot analysis of GFAP at 24, 48, and 72 h. β-Actin was the loading control. (**C**) Immunoblot analysis of whole LP DITNC1(ATCC) ± TNF (10 ng/mL, 48 h) cell lysates using anti-β_3_ Integrin and anti-Cx43 antibodies. β-Actin was used as a loading control. Values below the blots in (**B**,**C**) are the means ± s.e.m. of the ratio between the densitometric value of the first antibody signal and that of the respective β-Actin value (n = 3). (**D**) Representative images of low passage (LP) ± TNF pretreatment; Vimentin or Cx43 (green), F-actin (rhodamine-conjugated phalloidin in red), and nuclei (DAPI in blue) were visualized (scale bar = 50 µm). White arrows indicate Cx43 localized at the cell border. (**E**) Representative images of low passage (LP) ± TNF pretreatment (outlined rectangular area) and DITNC1(CH) astrocytes. Cells were treated with TRAIL-R2-Fc or stimulated with Thy-1-Fc, as indicated. FAs, F-actin, and nuclei were, respectively, visualized with vinculin staining (green), rhodamine-conjugated phalloidin (red), and DAPI (blue) (scale bar = 20 µm). (**F**) FA quantification. LP DITNC1(ATCC) cells were evaluated under untreated conditions (Control) or with TNF (10 ng/mL, 48 h) (LP + TNF), and DITNC1(CH) cells were stimulated with Thy-1 or treated with the negative control TRAIL-R2. The values in the graph are the means ± s.e.m. of >30 cells monitored per condition in each experiment (n = 3). # *p* < 0.05 compared to each respective control treated with TRAIL-R2. (**G**,**H**) Migration assay using LP DITNC1(ATCC) cells (**G**) or DITNC1(CH) cells (**H**) treated with TNF (+TNF) (10 ng/mL, 48 h) or left untreated (-TNF). The transwell assay was performed by adding 10^5^ cells in a serum-free medium containing Thy-1-Fc or TRAIL-R2-Fc to the top chamber for 2 h. Astrocytes that migrated to the bottom side through the insert pores were visualized by crystal violet staining and then counted. The graphs show the values normalized to the average of the control condition (LP + TRAIL-R2). At least six fields were counted per condition (n = 3) (mean ± s.e.m.). * *p* < 0.05; ** *p* < 0.01. (**I**) Cell proliferation was evaluated using the MTS assay in LP DITNC1(ATCC) cells treated with TNF (LP + TNF) or without TNF (LP) and in DITNC1(CH) cells after incubating them for 24, 48, and 72 h. The values in the graph indicate the optical density (O.D.) at 490 nm measured at the indicated time points (n = 3) (mean ± s.e.m.). *** *p* < 0.001 compared to LP DITNC1(ATCC) cells not treated with TNF at 72 h.

**Figure 3 ijms-25-09454-f003:**
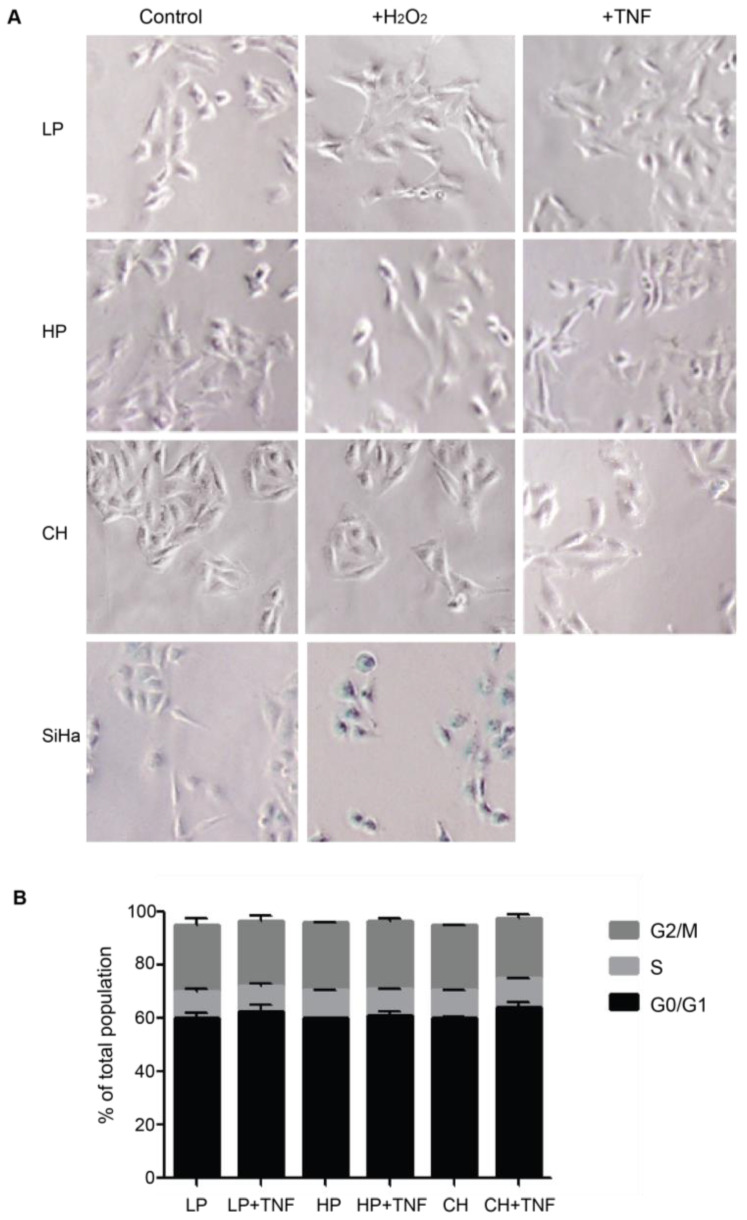
DITNC1 cells that undergo extensive passaging lack senescence-associated traits. (**A**) SA-β-galactosidase activity of cells treated with TNF or H_2_O_2_. Representative images of LP DITNC1(ATCC) (LP), HP DITNC1(ATCC) (HP), and DITNC1(CH) cells treated with 10 ng/mL of TNF or 10 µM H_2_O_2_ for 48 h. SiHa cells treated with 10 µM H_2_O_2_ for 48 h are also shown as a control for SA-β-galactosidase-positive cells. After incubation, SA-β-galactosidase activity was determined as described in Materials and methods. Magnification = 100×. Digital magnification= 3×. (**B**) Cell cycle assessment of TNF-treated cells. LP DITNC1(ATCC) cells treated with TNF (LP + TNF) or without TNF (LP), HP DITNC1(ATCC) cells treated with TNF (HP + TNF) or without TNF, and (HP) DITNC1(CH) cells treated with TNF (CH + TNF) or without TNF (CH). All TNF treatments were for 48 h. Cells were fixed and permeabilized with methanol, treated with RNAse, and stained with propidium iodide to determine the G0/G1 (black bars), S (light grey bars), and G2/M (dark grey bars) cell cycle stages by cell cytometry. The values in the graph are the means ± s.e.m. of three independent experiments.

**Figure 4 ijms-25-09454-f004:**
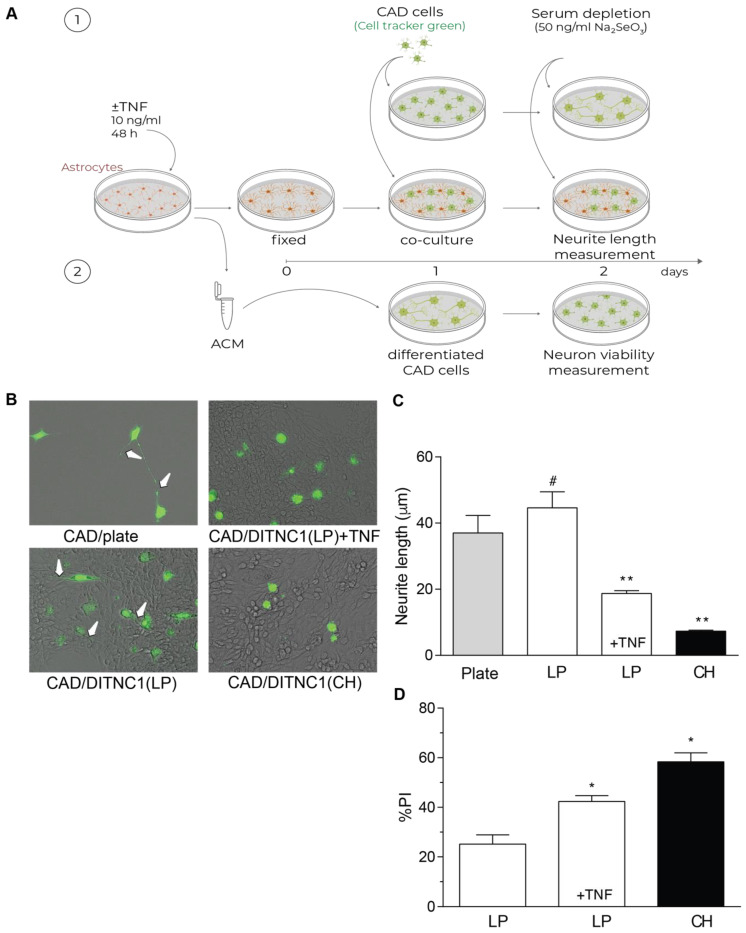
DITNC1 astrocytes with multiple passages inhibit neurite outgrowth and promote neuronal death. (**A**) 1. CAD cells (10,000 cells/cm^2^) labeled with Cell Tracker Green CMFDA (10 μM) were seeded onto a plate or co-cultured on top of a fixed monolayer of astrocytes [LP DITNC1(ATCC) cells with TNF (LP + TNF) or without TNF (LP), or DITNC1(CH) cells]. The extension of neuronal processes was induced by serum depletion and the addition of sodium selenite 50 ng/mL for 24 h. 2. Astrocyte-conditioned medium (ACM) was obtained from LP DITNC1(ATCC) cells with TNF (LP + TNF) or without TNF (LP) and DITNC1(CH) cells. Differentiated CAD cells were incubated with ACM for 24 h. (**B**) Representative microphotographs of fluorescent CAD cells (green) grown on a plate or over LP DITNC1(ATCC) cells treated with TNF (LP + TNF) or without TNF (LP) or over DITNC1(CH) astrocytes (bright field), obtained with a Cytation 3 instrument (BioTek, Santa Clara, CA, USA). White arrows indicate extended neurites. Magnification = 200×. (**C**) Quantification of neurite length (μm). For each quantification, the neurites of at least 50 cells were measured per condition using NeuronJ. The values in the graph are the mean ± s.e.m. (n = 3). Significant differences are indicated. ** *p* < 0.01, compared to control values on plates; # *p* < 0.05, compared to the value of LP + TNF. (**D**) Differentiated CAD cells were incubated with ACM for 24 h, and quantification of cell viability was then performed with propidium iodide (PI) staining. The values in the graph are the mean ± s.e.m. (n = 3). Significant differences are indicated by * *p* < 0.05 compared to LP control values.

## Data Availability

Any additional information required to reanalyze the data and reagents reported in this paper is available upon request to the corresponding author.

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
