# Peer review of "A Pro-Inflammatory Stimulus versus Extensive Passaging of DITNC1 Astrocyte Cultures as Models to Study Astrogliosis"

_ijms, 2024, doi:10.3390/ijms25179454_

Round 1
Reviewer 1 Report (New Reviewer)
Comments and Suggestions for Authors
Manuscript #: IJMS-3127603
Title: A pro-inflammatory stimulus versus extensive passaging of 2 DITNC1 astrocyte cultures as models to study astrogliosis
Authors: Perez et al.
In this study, the authors used immortalized rat neocortical astrocyte-derived cell line (DITNC1) and catecholaminergic neuron-like cell line (CAD) to study signaling mechanisms activated by syndecan-4/αvβ3 integrin/Thy-1 interaction. Further, the authors explored if astrocytes upregulate β3 Integrin after successive cell passages. Using DITNC1 cells with different passage numbers the authors studied reactivity markers by immunoblot analysis, immunofluorescence, and astrocyte adhesion/migration assays. The authors showed serial passage of cells mimicked the effects of inflammatory stimuli and elicited astrocyte reactivity. Importantly, they showed that astrocytes in late passages expressed and secreted factors that inhibited neurite outgrowth and were neurotoxic like reactive astrocytes in culture. Based on their observations, the authors describe two DITNC1 cell types, a non-reactive type that can be activated by tumor necrosis factor (TNF) and another that exhibits reactive astrocyte features even in the absence of TNF, thus demonstrating the importance of passage number in cell behavior. The authors present this as model to study astrocyte reactivity.
While the authors present important mechanistic information related to astrocytic reactivity, the following aspects need to be addressed.
1. For all assay methods (immunoblots, cytometry, migration, immunofluorescence, proliferation, senescence, cell cycle, neuronal outgrowth, and neuronal death), indicate how many experimental replicates (n) were included.
2. Immunoblots are depicted with an n=1 for various treatment groups (cell types or time points). The data is not convincing when only one sample is shown. The blots should be representative of replicates (at least an n=3 per treatment group). If replicate samples or blots have been run, they can be included in the supplemental information.
3. In immunoblots, it is not clear why different loading controls (Beta Actin or HSP90) are used in various immunoblot experiments. Some clarity is needed here.
Comments on the Quality of English LanguageMinor editing of English language required
Author Response
See point-by-point responses to reviewer N1 in attached document.

Reviewer 2 Report (New Reviewer)
Comments and Suggestions for Authors
It is an interesting study, but I have some questions and would like to share my opinions.
First of all, I understand the importance of presenting and emphasizing the continuity of the research, but I wonder if it is natural to introduce the laboratory in the abstract. The abstract should present the research objectives, results, and conclusions more concisely and clearly.
While I acknowledge that sufficient research has been conducted to compare the two research models, it seems to stop at merely comparing the models. It would be better if the significance and research value of each model were further elaborated.
In addition to discussing the significance and comparison of each model, it would be beneficial to describe their potential significance as disease models and mechanisms for future research. Moreover, the different research environments should be considered.
Although I implicitly understood and fully anticipated the risk of using old passaged culture models from an experimental perspective, it is meaningful that this has been experimentally demonstrated. However, it would be beneficial to add the future research implications of the model beyond merely proving the model itself.
The study results mention the importance of the passage number in cell behavior. Please indicate the clinical research significance of this number as a research model.
Furthermore, I also wonder whether the 100 passages condition can be linked to human aging conditions. Is there any evidence related to aging in the condition of 100 passages in aged mice?
Thank you.
Author Response
See point-by-point responses to reviewer N2 in attached document.

Round 2
Reviewer 2 Report (New Reviewer)
Comments and Suggestions for Authors
Thank you for your efforts in revising the manuscript
This manuscript is a resubmission of an earlier submission. The following is a list of the peer review reports and author responses from that submission.
Round 1
Reviewer 1 Report
Comments and Suggestions for Authors
Manuscript ijms-2220285
"The importance of establishing an appropriate model to study the reactivity process in astrocytes"
Authors: Leonardo A. Pérez, Jorge Díaz, Esteban Palacios, María Fernanda González, Samuel
Martínez-Meza, Ramón D. Pérez, Emanuel Jeldes, Ana María Avalos and Lisette Leyton
The manuscript is generally although interesting is not very pertinent because astrocytes can assume
many phenotypes, including among the “reactive phenotypes” and perhaps some markers indicated
in the manuscript like the expression of β3integrin may be interesting to include in the list of markers
of astrogliosis.
Major comments:
1- The aims of the stydy are not enterally clear and the title should be change. It seems that there
is a confusion between aging and the reactive phenotype that must be clarified.
2- DITNC1 (CH) seem to aged cells with a reactive phenotype. This means they are more
interesting to study aging than reactive gliosis, i.e they do not seem to be able to proliferate
as DITNC1-LP cells treated with TNFα.
3- The cell line model DITNC1 (CH) and other versions of DITNC1-LP stimulated with TNFα
do not quite represent all the features of reactive astrocytes, which may differ according to the
brain region affected as well as the triggers of neuroinflammation that change with the
pathologies of CNS involved.
4- According to your results both lines DITNC1 (CH) and DITNC1-LP and treated with TNFα
present neurotoxic effects, by inhibiting neurite growth. However, reactive astrocytes not
always have neurotoxic effects. In this case it would be interesting to identify which reactive
phenotype these reactive astrocytes have (A1 or A2) and the cytotoxic factor involved
inhibition of neurite outgrowth and neuronal death.
5- According to what is known, astrocytes may present diverse functions according to brain
region (see Khakh and Sofroniew 2015; doi:10.1038/nn.4043) and reactive astrocytes may
also present different phenotypes and functions depending on the interaction with microglia
and the neuropathology: A1 and A2 (see Barres Ben et al. 2017, doi: 10.1038/nature21029;
and doi: 10.1016/j.celrep.2017.04.047; doi: 10.1186/s40035-020-00221-2). This information
should be taken into account in the discussion.
6- I also would be interesting to see the influence of other cytokine release by microglia in the
reactivity and response of these so called “cell models of reactive astrocytes”.
Manuscript ijms-2220285
"The importance of establishing an appropriate model to study the reactivity process in astrocytes"
Authors: Leonardo A. Pérez, Jorge Díaz, Esteban Palacios, María Fernanda González, Samuel
Martínez-Meza, Ramón D. Pérez, Emanuel Jeldes, Ana María Avalos and Lisette Leyton
The manuscript is generally although interesting is not very pertinent because astrocytes can assume
many phenotypes, including among the “reactive phenotypes” and perhaps some markers indicated
in the manuscript like the expression of β3integrin may be interesting to include in the list of markers
of astrogliosis.
Major comments:
1- The aims of the stydy are not enterally clear and the title should be change. It seems that there
is a confusion between aging and the reactive phenotype that must be clarified.
2- DITNC1 (CH) seem to aged cells with a reactive phenotype. This means they are more
interesting to study aging than reactive gliosis, i.e they do not seem to be able to proliferate
as DITNC1-LP cells treated with TNFα.
3- The cell line model DITNC1 (CH) and other versions of DITNC1-LP stimulated with TNFα
do not quite represent all the features of reactive astrocytes, which may differ according to the
brain region affected as well as the triggers of neuroinflammation that change with the
pathologies of CNS involved.
4- According to your results both lines DITNC1 (CH) and DITNC1-LP and treated with TNFα
present neurotoxic effects, by inhibiting neurite growth. However, reactive astrocytes not
always have neurotoxic effects. In this case it would be interesting to identify which reactive
phenotype these reactive astrocytes have (A1 or A2) and the cytotoxic factor involved
inhibition of neurite outgrowth and neuronal death.
5- According to what is known, astrocytes may present diverse functions according to brain
region (see Khakh and Sofroniew 2015; doi:10.1038/nn.4043) and reactive astrocytes may
also present different phenotypes and functions depending on the interaction with microglia
and the neuropathology: A1 and A2 (see Barres Ben et al. 2017, doi: 10.1038/nature21029;
and doi: 10.1016/j.celrep.2017.04.047; doi: 10.1186/s40035-020-00221-2). This information
should be taken into account in the discussion.
6- I also would be interesting to see the influence of other cytokine release by microglia in the
reactivity and response of these so called “cell models of reactive astrocytes”.

Reviewer 2 Report
Comments and Suggestions for Authors
The paper “The importance of establishing an appropriate model to study the reactivity process in astrocytes” by Pérez et al. proposed a cell model to study astrocytes reactivity indicating some specific markers that can be an index of the astrocytes stage. In particular, referring to the cell model, the authors suggest an ageing/senescent astrocytes culture related to the number of the passages in cells culture that should shift from naive (no-reactive to the reactive state) mimicking an ageing situation. The authors used both immortalized DITNC1 cells (with a number of passages >100) and DITNC1 cells subjected to or a great number of passages (>90) or to a low number. They show that long-term cultured or aged DITNC1 astrocyte cell lines have more β3 Integrin at the cell surface, and migrate when exposed to Thy-1 similarly to the immortalized cells.
To the other hand, LP DITNC1(ATCC) cells, with only few passages give rise to non-reactive astrocytes but when treated with pro-inflammatory agents showed reactivity, increased adhesion, migration, and proliferation.
This is an interesting study on methodological aspects which adresses an important issue on what cell model can be useful, and therefore “clinically relevant”, in studying astrocytosis processes. above all in the view, to obtain comparable data from different labs
Despite the important questions developed in this paper a critical point has been revealed.
The authors affirm that their study demonstrates that astrocytes cultures stressed with several passages undergo changes in cell morphology but unfortunately this aspect is not evidenced by microscopy images.
This assertion is based on the greater or little expression of some markers that could be index of changes in astrocytes but only matching this result with an immunohistochemical analysis can be appreciate and confirmed.
The only image is confined in figure 2C but it takes in consideration Low passage with TNF pretreatment , and DITNC1(CH) astrocytes, treated with TRAIL-R2-Fc or stimulated with Thy-1-Fc and marked with FAs, F-actin and nuclei were respectively visualized with vinculin staining (green), Rhodamine-conjugated from which is not possible to evidence a morphological change.
Representative images on astrocytes morphology (i.e. GFAP, phospho-vimentin (pVIM), …) is desiderable to confirm the reactivity state.
From an overall analysis of the text, the manuscript is clear and well-written.
Reviewer 3 Report
Comments and Suggestions for Authors
The presented results of the study are interesting and new. The article is framed in accordance with the rules of the publisher. Of the small methodological comments, it is proposed to expand the introduction, as well as to give an experimental scheme for a better perception of the material.
Reviewer 4 Report
Comments and Suggestions for Authors
The manuscript by Leonardo A. Perez and colleagues presents a study on changes in markers of gliosis in low and high passage astrocytes. The manuscript suffers from critical conceptual and experimental deficiencies as detailed below.
(1) The beginning of the manuscript sounds like a lab advertisement in the Abstract section “Our laboratory….and connexin43”. Around 25% of the references are self-citations. While not necessarily wrong or unsound, readers of the manuscript might get an impression that the manuscript is a self-appreciating piece of work.
(2) The biggest conceptual problem here is that the entire manuscript in general and the 3rd and 4th paragraph of the Discussion section (page 12) in particular seem to be blending the concept of chronological aging with that of in vitro aging due to multiple passaging. The biochemical, cell biological, and molecular biological changes associated with chronological aging vs in vitro aging might have some overlapping components, however, two are distinctly different processes. The two, therefore, can not be synonymized, particularly in light of the fact that in vitro aging is associated with cancerous (immortal) cells (accumulating mutations with every passage) while chronological aging is associated with normal cells (often in context of whole tissue or organism; associated with robust mechanisms to prevent abnormal ploidy, DNA damage, and mutations etc.). Most of the symptoms of the in vitro aging observed in multi-passage cells are due to accumulation of chromosomal aberrations and mutations to the point that original identity of the cell line is lost beyond certain passage number. Until unless evidence is provided that there are no chromosomal/DNA changes in HP DITNC1(ATCC) and DITNC1(CH) cells compared to the LP DITNC1(ATCC) cells used in this study, changes observed in former two cell lines in this study can not be attributed merely to aging. Equating in vitro aging observed in HP DITNC1(ATCC) and DITNC1(CH) cells with chronological aging is, therefore, an unsubstantiated proposition.
(3) Similarly, paragraph 4 on page 2: "under inflammation...with time" seem to imply that astrogliosis and astrosenescence are the same process. This sentence suffers the same issue as indicated in point (2) above.
(4) The conclusion stated in section 3.1 "Aged, but not....inflammatory stimuli" is not unequivocally supported by the data presented in Fig 1 (see below):
(4 a) Fig 1A: Both LP DITNC1 and HP DITNC1 cells shape, size, number, free space appears to be very similar contradicting the claim made on page 6 in section 3.1 line 8--11 "At low passages, the cell...between clusters".
(4 b) Fig 1B, C3d levels are same for LP DITNC1 and HP DITNC1 cells, again indicating very similar nature of LP and HP cells.
(4 c) Fig 1B, loading control HSP90 not same in all cases, hence a fair comparison can not be made for β3-integrin levels for LP, MP, HP and CH. Subsequent plot of β3-integrin/HSP90 ratio is irrelevant due to this fact. Hence, claims based on this fig on page 6 in section 3.1, lines 16--28 "HP DITNC1 (ATCC)....DITNC1(CH) cells (Fig. 1B), remain disputed.
(4 d) Fig 1C, histogram indicates insignificant difference between LP DITNC1 and HP DITNC1 cells. This shakes the claim made on page 6, section 3.1, lines 30--32 "levels of b3 integrin...cdll (Fig 1c, 1D)".
(4 e) Fig 1E, it is not clear with respect to what the fold change was calculated. HP DITNC1 in TRAIL-R2 case seems to be migrating less than control, while DITNC1(CH) does migrate significantly compared to LP DITNC1. Again, it suggest that HP DITNC1 and DITNC1(CH) are completely different beasts, again putting question mark on authors' hypothesis equating HP DITNC1 and DITNC1 (CH) cells.
(5) Similarly, the conclusion made in section 3.2 "Cell with a low ... treated with TFN" are also not well substantiated by the data shown in Fig 2:
(5 a) Fig 2A, The loading control β-actin levels are not similar in all blots indicating the differences in protein loading which might give rise to differences in the intensities of GFAP band at different time-points. Hence, the data can not be trusted, unless supplemented by RT-PCR based readout for GFAP expression at these time-points. The experiment is flawed is also supported by the fact that GFAP level at 72 h becomes almost equal to non-treated levels.
(5 b) Fig 2B, blot quality of β3 integrin is not good enough to conclude anything from it. The blot for Cx43 again suffers from unequal loading problem indicated by varying β-actin band, hence, conclusion on page 8, line 30 "As shown in ...TNF treatment" remain less than convincing.
(5 c) How the GFAP and Cx43 levels compare with HP DITNC1 and DITNC1(CH) cells, has not been shown. Neither has been shown the concentration-dependence of these markers on TNF concentration.
(5 d) In Fig 2C, vinculin staining is much higher in LP+Thy-1 compared to CH+Thy-1, while in Fig 2D, number of FA/cell are almost same for LP and CH in Thy-1 case with very small SEM values. why is this discrepancy between these two read-outs?
(5 e) In Fig 2E,F, it is not clear how long TNF pre-treatment was given to the cells.
(5 f) The statement "Here, although LP DITNC1(ATCC)....obtained at 24 and 48h" on lines 45--47 of page 8 is misleading as Fig 2G clearly shows OD value increasing from ~0.25 at 24h to ~0.45 at 72h for these cells. This indicates a significant increase in cell number and hence can not be interpreted as insignificant. Therefore, conclusion connecting proliferation with reactive astrocyte state only in TFN freated cells (lines 49--50 on page 8) is not valid.
(6) Fig S1 quality is not good and different panels seem to have different brightness/contrast settings making the comparison very difficult.
(7) It is not clear at what time-point of cell growth the data in Fig S2 was recorded. Fig 2G suggests significant proliferation activity in TNF-treated LP cells as well as CH cells, while Fig S2 shows all cells in the same cell cycle state. The two Fig are thus contradictory to each other. There has to be difference in the cell cycle states between rapidly proliferating cells and non-proliferating cells unless there is something wrong with the experiment or data analysis.
(8) In Fig 3A, please show the images of CAD on LP DITNC1 and HP DITNC1 in order to reveal the neurite growth features. In addition to that, the support cell monolayer should be labeled with another dye to make sure that the distribution of CAD vs DITNC1 is uniform across different DITNC1 types. In the current Fig 3A it is difficult to conclusively say if the differences in neurite length is due to underlying cell type (plus TNF) or just due to the voids in the monolayer/non-uniform coverage. Due to these potential pitfalls, the inferences made on page 10, lines 29--35 remain disputed.
(9) Page 12, line 40 mentions "undergoing a reversible stage". There is no data in the manuscript supporting the reversibility of these astrocytes.